# A Systematic Literature Review of Soft Skills in Information Technology Education

**DOI:** 10.3390/bs14100894

**Published:** 2024-10-02

**Authors:** Farhad Sadik Mohammed, Fezile Ozdamli

**Affiliations:** 1Department of Computer Information Systems, Near East University, Lefkoşa 98010, Turkey; 20204542@std.neu.edu.tr; 2Department of Management Information Systems, Near East University, Nicosia 99138, Turkey

**Keywords:** soft skills, higher education, IT education, industry

## Abstract

This research addresses the importance of the soft skills approach, which encompasses problem-solving, collaboration, interpersonal and communication skills for higher education in the information technology (IT) field. IT graduate students continue to face difficulties in meeting the employability criteria of the global information technology sector due to mismatching capabilities, such as the discrepancy between the technical knowledge obtained in academia and the practical skills expected by employers. This systematic literature review used PRISMA guidelines for data collection. Papers were examined using the inclusion–exclusion criteria, which included concentrating on full-text studies about soft skills in higher education published in English between 2018 and 2024. The keywords used by the inclusion and exclusion criteria are soft skills, higher education, university, undergraduate, graduate, IT, information technology, software, computer science, programming, information systems and IS. The SCOPUS search engine platform found 2088 documents, and the (WOS) database obtained 1383 documents. To comprehend the significance of soft skills in the field and its effect on graduates’ employability, 69 papers were carefully examined. The rapid change following Industrial Revolution 4.0 has transformed the working environment, challenging new IT graduates to be competent in the working environment. This study highlights the importance of soft skills and self-awareness in university education, revealing that current curricula must adapt to the rapidly changing job market, especially post Industry 4.0. The literature review indicates that despite high technical competence, graduates lack essential soft skills like communication, teamwork, and problem-solving, creating a gap between new graduates and industry expectations. Hopefully, this study’s results will contribute to understanding the functionality and necessity of soft skills in the behavioral sciences literature. To bridge the skills gap between industry demands and technical proficiency, academic institutions should incorporate creative teaching approaches prioritizing soft skills like problem-solving, teamwork, and communication. Universities, recent graduates, and companies must work together to modify courses to meet the needs of a job market that is changing quickly.

## 1. Introduction

Traditional learning methodologies in higher education (HE) are at odds with the changing workforce requirements. Numerous studies show that university graduates lack important abilities such as interpersonal communication, social skills, creativity, and project execution [1,2]. This underlines the critical importance of developing 21st-century skills for global economic involvement and professional competence. According to Draaijer et al., the graduation averages incorrectly forecast students’ future professional success in work placements [3,4]. Employability is emphasized in higher education because of the possible economic risks associated with rising unemployment rates, a common concern in developed and developing countries [5,6]. The literature divides learned skills into cognitive and non-cognitive (soft skills). Non-cognitive skills are attitudes that affect how individuals approach learning and interact with people around them [7]. As stated by other researchers, the most prominent features of soft skills are emotional awareness, positivity, interaction, people management, conflict management, strategic thinking and fast learning skills. In contrast, numerical, language and data literacy skills are not specific to soft skills [8].

Emotional intelligence is an important skill that forms the basis of soft skills. Emotional intelligence’s self-awareness and self-management components contribute positively to skills such as leadership and problem-solving [9]. Thanks to their emotional intelligence, individuals working in technology can communicate better, increase their problem-solving abilities, and be more effective in innovative thinking. For example, major technology companies such as Google and Microsoft view emotional intelligence as an important competency for leadership and team management [8].

Emotional soft skills are critical for success in the workplace and social life because they improve individuals’ ability to cope with stress, adapt to change, and communicate effectively in complex social environments [10]. These skills enable employees to be more productive and adaptable, especially in the technology sector, which requires intense workloads and complex project management. The Organization for Economic Cooperation and Development (OECD) has also developed a framework stating that social–emotional skills are critical in cooperation, task performance, emotional regulation, and open-mindedness. According to this framework, sub-skills such as tolerance and sociability are also included in these areas. The literature shows that emotional skills are important in academic success, career-related decision-making self-efficacy, and rapport with colleagues [11].

In addition, another concept that comes to mind when soft skills are mentioned is emotional intelligence. Especially in technology, emotional intelligence is the key to success [12]. Individuals with developed emotional intelligence can collaborate comfortably and have the skills to be good leaders. Individuals who know and manage their emotions can make good choices and perform better. If employees in the field of technology can remain calm and positive, a more comfortable working environment can be realized. In addition, stronger teams are established by bringing together individuals with empathy skills. Empathy plays a critical role in conflict management and adaptation in technology teams. Individuals with developed empathy skills strengthen communication within the team and can produce more creative solutions for projects. It increases the cooperation of teams and creates a stronger and more productive working environment [13]. The Adults Online Course platform training emphasized that “Emotional intelligence is as important as technical skills in the technology sector”.

Bhat and Gupta state that critical thinking, communication, creativity and collaboration (4Cs) are soft skills and are complex and vital factors affecting employers’ productivity. The researchers also emphasized that the soft skills that are increasingly scarce in the workforce are “compassion and understanding”, “feedback”, “motivation”, “collaboration synergy”, “practical knowledge”, “interpersonal skills”, and “team culture” [14]. Soft skills are personal qualities, habits, attitudes and social courtesies that make individuals good employees and suitable for work. Also, soft skills are personal qualities that improve individuals’ interaction skills, job performance and career prospects. In other words, subtle behaviors and communication styles make managing a work environment or interacting with another person easier [15].

In today’s higher education, the system often focuses on students learning theoretical knowledge, while the practical application of the learned knowledge is sometimes overlooked. For this reason, students cannot effectively use the knowledge they have learned when they start their professional lives. The education system should not only transfer knowledge but also practice using this knowledge.

Two hundred academics and experts participated in the “Building Tomorrow’s Talent” study sponsored by Workday. According to the results of this study, employers care more about interpersonal skills than the GPA on the transcript. Some universities have started to indicate the skills of their graduates on their transcripts [16]. Similarly, Vleuten from Maastricht University stated that students’ inability to apply what they learn in practice is a fundamental problem of education.

Another study in Finland illustrates that the impact of generic skills in education is challenging [17]. Therefore, the various academic programs require a specific model design. Moreover, the necessity of an analytical self-assessment process is vital. Tsortanidou states that collaboration, problem-solving skills, and creativity are among the skills targeted in the twenty-first century [18]. Furthermore, Gretter and Yadav have stated that for individuals to acquire these skills, students should be creators rather than gaining specific knowledge [19]. It is specified that improving these skills should be included in the current culture. Graduates’ lack of soft skills seems to be the fundamental reason causing unemployment. The unemployment of new university graduates can cause social distress and psychological complications in the long term, as stated by researchers [20].

Many researchers state that IT workers do not have the soft skills required by the digital age as much as they desire and, therefore, have difficulty joining the workforce [21,22,23]. Other researchers also state that no matter how good the technical knowledge and competencies are, the lack of sufficient soft skills creates incompatibility with changing job demands and duties [21,24].

In Hong Kong, IT has been declared a priority development industry. In addition, a business field called “new collar” has been defined, and it is stated that technical knowledge and soft skills must be at a good level in this field [25]. In a statement made by the University of Pennsylvania, it was specified that students who study computer science in the learning process with new learning approaches could be successful if they have skills such as time management, a willingness to cooperate, being determined, and not being afraid to seek for help [3].

A fundamental role of computer scientists is to produce solutions through collaborative problem-solving with programmers, IT experts, and mechanical or software engineers [26]. However, Flavin emphasizes in his report on necessary computer science abilities that technical knowledge alone is insufficient, emphasizing soft skills considerably [27]. These critical success factors include communication, teamwork, presentation, self-awareness, and professionalism, linked to essential personality traits and abilities developed by experience and practice. Another study by Grover and Pea argues that programming goes beyond code to include self-expression [28]. Furthermore, Varela states that individuals must have adaptive, change-sensitive problem-solving skills to succeed in changing global settings, including software and hardware creation [29]. World-renowned organizations, such as the OECD and UNESCO, are actively exploring the skills required for 2030. These organizations demand that higher education institutions (HEIs) support national, international, and industrial curriculum reforms that promote quality, flexibility, and lifelong learning from a sustainability standpoint.

In a study conducted by Dobslaw et al. in Sweden in 2023, they performed a trend analysis of technology in job postings over the last six years. As a result of the study, they revealed that the demand for cloud and automation technologies such as Kubernetes and Docker has increased in job postings. Still, these topics are not sufficient in higher education curricula. The researchers stated that there are differences between higher education curricula and job postings, that while concepts are emphasized more in universities, technologies are mentioned in the IT industry. They suggested that future studies should include research that includes soft skills [30].

The impact of soft skills development on the workplace is multifaceted, especially for IT professionals [31]. Effective communication and determining solutions in IT projects depend on teamwork and cooperation. Teams that collaborate effectively also increase customer relationships, thus increasing customer satisfaction [31,32]. As seen in the literature, changes in the information technology (IT) field increase the importance of soft skills. Although technical skills are indispensable in the IT industry, it is understood that qualified soft skills are equally important for individual and team success [33]. At the same time, work environments that encourage creativity enable the development of new technology and increase efficiency in business processes [34,35]. In addition, a high level of soft skills increases employees’ motivation, controls stress management and improves social relations.

This study aims to investigate the role of soft skills in IT higher education and assess its impact on the employability of graduates in the IT industry. With an increasing demand for proficient and well-equipped IT professionals, universities face pressure and challenges to produce graduates with relevant skills. However, mismatches in skills and a lack of soft skills have become barriers for IT students entering the employment workforce. We recognize the potential of soft skills to bridge this gap and mitigate the unemployment rate. This research identifies trends in soft skills elements within studies conducted for information technology students.

As mentioned above, soft skills are one of the determining factors for the success of IT employees. The changing dynamics of the IT industry expect employees to have both technical knowledge and practical knowledge, collaboration, time management and communication skills at a sufficient level in soft skills. In higher education institutions, it is seen that these demands are not sufficiently focused on meeting, and graduates have difficulties in their professional lives. For this reason, examining the research conducted on these skills in higher education institutions’ curricula is important from an industrial and academic perspective. This review identifies a gap in the literature and a lack of comprehensive studies combining and analyzing the results in this context. The research questions guiding this investigation are as follows:Which specific soft skills have been the primary focus in studies developed for students undergoing IT education?What challenges or difficulties have been encountered in developing and implementing soft skills studies for IT students?What recommendations or suggestions emerge from studies focusing on developing soft skills for students in IT education programs?What are the emerging trends in fostering soft skills in IT education?

These research questions aim to comprehensively recognize the impact, challenges, and recommendations associated with soft skills in information technology education.

## 2. Research Methodology

Studies that evaluate research found in the literature are called review articles. In review articles, the relevant literature is carefully identified and synthesized. Review articles provide future researchers with an advanced understanding of the topic and help identify research gaps. There are review studies based on different methods, including bibliometric reviews, narrative reviews, scoping reviews, systematic literature reviews, and meta-analyses [36]. In cases where it is impossible to obtain homogeneous data sets across different types of studies, systematic literature studies using the meta-analysis method are more appropriate for qualitatively evaluating general trends and theoretical frameworks.

The study was planned to answer the determined research questions using PRISMA 2020 (Preferred Reporting Items and Systematic Reviews and Meta-Analysis), which is widely used in systematic literature reviews [37].

A systematic literature review is the process of identifying and critically evaluating the studies in the literature on the subject, as well as collecting and analyzing data from the identified studies [37]. Since the inclusion and exclusive criteria are determined in advance in the systematic literature review, bias in the results can be kept to a minimum [38].

### 2.1. Search Strategy

Scopus and WOS are common search engines used for academic literature searches in terms of the coverage of scientific journals, conference proceedings, and books in the peer-reviewed literature [39]. In addition, Scopus and WOS have become preferred databases because they mostly list journals with high-impact factors and have an understandable and simple interface [40].

To achieve the objectives of the systematic literature review, the appropriate classification was evaluated and categorized according to the context. The study’s authors searched the databases by determining a set of keyword combinations. This query technique uses Boolean logic, which uses keywords and phrases related to the topic with the combination operators AND, OR, and NOT to eliminate the wide area of the topic by a certain period.

The research keywords were determined according to the given criteria: “Soft skills”, “higher education”, “university”, “undergraduate”, “graduate”, “IT”, “Information Technologies”, “software”, “computer science”, “programming”, “information systems”, and “IS”. The research conducted with these keywords includes those published between 2018 and 2024 that are open access and in English. The identified keywords were scanned in the “Title-Abs-Key” field of the SCOPUS database and the “Topic” field of the WoS database. These fields perform the scan, which includes the title, the abstract and the keywords of the studies.

WOS and Scopus have easy-to-use filtering features. The filters have been made by the inclusive and exclusive criteria using the formula given below:
$$\begin{array}{l} (\mathrm{TITLE-ABS-KEY}(“\mathrm{Soft skills}”)\;\mathrm{AND TITLE-ABS-KEY}\;((“\mathrm{higher education}”\\ \mathrm{OR}\;“\mathrm{university}”\;\mathrm{OR}\;“\mathrm{undergraduate}”\;\mathrm{OR}\;“\mathrm{Graduate}”))\;\mathrm{AND TITLE-ABS-}\\ \mathrm{KEY}\;((“\mathrm{IT}”\;\mathrm{OR}\;“\mathrm{Information Technologies}”\;\mathrm{OR}\;“\mathrm{Software}”\;\mathrm{OR}\;“\mathrm{Computer}\\ \mathrm{Science}”\;\mathrm{OR}\;“\mathrm{Programming}”\;\mathrm{OR}\;“\mathrm{Information Systems}”\;\mathrm{OR}\;“\mathrm{IS}”)))\;\mathrm{AND}\\ (\mathrm{LIMIT-TO}(\mathrm{OA},\;“\mathrm{all}”))\;\mathrm{AND}\;(\mathrm{LIMIT-TO}\;(\mathrm{PUBYEAR},\;2024)\;\mathrm{OR LIMIT-TO}\\ (\mathrm{PUBYEAR},\;2023)\;\mathrm{OR LIMIT-TO}\;(\mathrm{PUBYEAR},\;2022)\;\mathrm{OR LIMIT-TO}\\ (\mathrm{PUBYEAR},\;2021)\;\mathrm{OR LIMIT-TO}\;(\mathrm{PUBYEAR},\;2020)\;\mathrm{OR LIMIT-TO}\\ (\mathrm{PUBYEAR},\;2019)\;\mathrm{OR LIMIT-TO}\;(\mathrm{PUBYEAR},\;2018))\;\mathrm{AND}\;(\mathrm{LIMIT-TO}\\ (\mathrm{DOCTYPE},\;“\mathrm{ar}”))\;\mathrm{AND}\;(\mathrm{LIMIT-TO}\;(\mathrm{LANGUAGE},\;“\mathrm{English}”))\;\mathrm{AND}\\ \left(\mathrm{LIMIT-TO}\;(\mathrm{SRCTYPE},\;“j”)\right) \end{array}$$

### 2.2. Searching Criteria and Variables

The inclusion and exclusion criteria narrow the broader topic to a more reliable context, emphasizing the study’s credibility and validity. Table 1 displays all the variables included and excluded in this study.

The database search selection has commenced with 3471 articles. Then, two thousand eighty-eight articles were found in the SCOPUS database search engine. Also, one thousand three hundred eighty-three papers were found on the Web of Science, a published papers collection search platform. The journal publications were checked from 2018 to the present date. Nevertheless, many papers were excluded following PRISMA’s examination process for several reasons, such as duplicated papers, books, conference proceedings, reports, and magazines. Also, the open access articles are included. The language was filtered to English to avoid wasting time translating (Figure 1).

Following the inclusion and exclusion criteria, 243 articles were accessed in WoS, and 308 published articles were found in the SCOPUS search engine platform. According to the PRISMA process, 175 duplicated papers were eliminated (duplicates were removed using the Mendeley 1.19.18 software’s basic functions). It followed the screening process of abstracts, of which 257 papers were excluded, since it was not related directly to the subject area of the study. Then, from the eligibility phase of the PRISMA process, another 15 papers were omitted because of the different language publications, and 35 full papers were eliminated because the results were not fit for the research questions. The total number of articles included in this systematic literature review is 69.

Both authors independently assessed abstracts and titles for inclusion and exclusion criteria. Two authors extracted data independently using a standard Excel form for the risk of bias and applicability assessment. The writers’ concordance index (Cohen’s Kappa Coefficient) was 0.91.

## 3. Literature Review

Journal papers from the last six years, 2018–2024, were examined. The table below presents the themes of soft skills obtained from the articles included in the study’s scope (Table 2).

## 4. Results

The following arguments and discussions of this systematic literature review results are based on the research questions. This literature review results illustrate a growing concern about the mismatch between the new educational program, employment requirements, and working industry guidelines.

### 4.1. Significant Soft Skills Emphasized in IT Education

The study of soft skills in IT education reveals several interests. Various approaches and strategies have stressed the importance of intra-personality, teamwork, communication, and problem-solving abilities.

Soft skills are becoming increasingly important in IT education, molding students’ competencies and ensuring their readiness for a dynamic labor market. This extensive analysis explores numerous researchers from worldwide focusing on developing and integrating soft skills in IT education. Studies in IT education have repeatedly revealed a wide range of soft skills deemed necessary for IT students. Based on the findings of the data compilation, specific soft skills that have been recurring crucial topics in this research include:

Communication skills: A consistent emphasis in many studies has been placed on communication skills. Researchers have assessed intra-personality, impression control, and interpersonal competencies in an engineering environment using the Business-focused Inventory of Personality (BIP) [82].

Teamwork and cooperation: This vital element has been identified as a key ability for information technology professionals. Researchers used the Multinational, Intercultural, Multidisciplinary, and Intensive (MIMI) technique to help computer science students improve their teamwork skills [62].

Problem-solving and critical thinking: Problem-solving and critical thinking abilities are frequently highlighted. Other research studies stress the relevance of problem-solving in the educational preparation of IT students and the essentials of the critical thinking process [50,71].

Adaptability and flexibility: The IT industry’s dynamic nature needs adaptability and flexibility. Other researchers emphasize the importance of students adapting to changes and demonstrating flexibility in their approaches [83,84].

Leadership skills: Leadership abilities have been identified in various investigations. The authors emphasized the necessity for hard and soft skills, especially leadership, for Thai vocational students [85]. There is a strong correlation between university education skills taught in enhancing students’ leadership skills, self-confidence and decision-making in the job environment [58].

Intrapersonal skills and self-awareness: Intrapersonal skills are recognized, including self-awareness. Researchers developed a theoretical hybrid model with AI features to measure students’ soft skills [86].

Ethics and moral values: This section investigates the ethical dimensions of soft skills. Zabidi et al. explored the incorporation of ethical and moral principles into educational institutions, focusing on the socio-cultural impact on skill development [87].

Creativity and innovation: It is considered a highly valued quality skill. Game-based learning (GBL) was used in studies to improve students’ inventiveness and creative thinking [88,89].

Socio-emotional skills: Socio-emotional skills are highlighted, especially emotional intelligence. During the COVID-19 pandemic, teaching and learning in South American countries emphasized the need for self-awareness skills [90]. Also, emotional engagement is crucial and positively impacts IT students to demonstrate tasks online as digital learning becomes a phenomenon [65].

Self-control and time management: Both elements are measured as a necessity, because of the importance of active learning, communication, collaboration, creativity, and critical thinking [91].

Motivation and curiosity: Motivation and curiosity are encouraged for continual learning. Researchers investigated skills supplied to employers for fresh graduate skill competencies, focusing on willingness and analytical analysis [92]. Individual students from the IT field must expect some requirements of personally guided experience [93].

Business and entrepreneurial Talents: Researchers emphasized the importance of partnership between students and industry through an entrepreneurial program [94]. Theoretically, outstanding IT students with managerial skills are capable and crucial conditions for the performance of the company when they are employed, as stated by [57]

Interpersonal skills: Interpersonal skills are handled constantly in the literature. Researchers examined employers’ perceptions of graduate students’ soft skills, specifically interpersonal, problem-solving, moral, ethical, self-management, and thinking abilities [95].

Cognitive skills: Skills such as critical thinking and analytical cognition are continuously emphasized. Lin and Yo created a theoretical framework to evaluate undergraduate courses by incorporating characteristics of soft skills such as leadership, communication, and reasoning [96]. In the current digital era of teaching, recognizing the factors that influence learning to motivate students is vital [66].

Willingness to Learn: The common theme is the enthusiasm to learn. Ref. [69] investigated the soft skills required by Malaysia’s engineering industry, concentrating on interpersonal, intrapersonal, and business abilities.

These results highlight the heterogeneous yet linked nature of soft skills in IT education, underlining their critical role in developing well-rounded and adaptive IT professionals. As detailed in this comprehensive report, the investigation of soft skills in IT education demonstrates a global understanding of their growing importance in developing well-rounded and flexible professionals for the dynamic labor market. The highlighted soft skills range from communication and teamwork to problem-solving and leadership, forming diverse yet interrelated aspects. These are critical for success in the ever-changing IT business. Including empirical evidence from several studies strengthens the legitimacy of these assertions. Although, focusing on practical approaches, such as game-based learning and the MIMI technique, provides a tangible dimension to the discussion.

Furthermore, other factors included ethical components and socio-emotional skills in a worldwide pandemic era. Also, the multifaceted nature of self-control and time management emphasizes the nuanced and holistic approach required for soft skill development. This list reflects the current status of soft skills in IT education. However, it acts as a valuable resource for educators, students, and industry stakeholders navigating the complex environment of educating IT workers toward success.

### 4.2. Exploring Barriers to IT Students’ Development of Soft Skills

The difficulties experienced in soft skills studies for IT students are multifaceted and can be summarized as follows:

Communication and relationship-building challenges: The Business-focused Inventory of Personality (BIP) model revealed that IT students, particularly engineering students, struggle to develop personal relationships and communicate effectively [82]. According to researchers, 90% of CS/IT students lack the problem-solving and critical analytical-thinking skills necessary for effective communication and relationship-building [71]. The study conducted by Caggiano and his colleagues stated that while information technology engineering graduates had sufficient technical skills, they had difficulty with social and communication skills [42]. This result, which emphasizes the importance of emotional skills in one’s professional life, shows that employees need to develop not only technical skills but also social skills that include emotional skills.

The gap between industry expectations and university education: According to researchers, while industry and employment believe IT graduates are well-equipped with technical abilities, the transition to work frequently fails due to a lack of the life-long learning of soft skills [47]. The absence of collaboration between colleges, universities, and the IT sector has hindered the understanding career requirements following graduation [50].

Teamwork and gender differences: Researchers observed difficulty developing group assignments in database administration due to gender inequalities, which impacted teamwork experiences [47].

Entrepreneurship skills and integration: Adding entrepreneurship skills to the school curriculum is difficult because it is based on individual traits [97]. However, in higher education institutions (HEIs), students can learn about entrepreneurship.

Impact of COVID-19 on soft skills gap: The soft skills gap has widened since the COVID-19 pandemic because of fast-changing surroundings, which cause IT students to be oblivious to industry developments [98].

Resistance to curriculum redesign: Including soft skills in the IT curriculum may necessitate program modification and additional work to attain learning outcomes in core course modules [50].

Difficulty in online soft skills teaching: Due to demographic and technical considerations, applying a soft skills approach to online teaching and learning post-COVID-19 is difficult [43]. Distance learning processes can create difficulties in terms of the development of students’ emotional skills such as self-regulation and motivation. In order to overcome these difficulties, activities that support emotional intelligence should be included. A combination of face-to-face and eLearning can minimize the gap of soft skills issues which prepare classes that students can engage and interact with independently and be motivated toward learning, as stated by [81].

Industry blaming educational institutions: The industry accuses higher education institutions (HEIs) of failing to prepare students with the requisite soft skills, resulting in a lack of initiative [99].

Challenges in developing countries: Low-level soft skills in underdeveloped nations such as Bangladesh are difficult to promote due to social inequality, the digital divide, and a lack of initiative by education officials [83].

Hard skills are prioritized over soft skills: Despite the industry’s emphasis on soft skills, there has been little effort to incorporate them into educational programs [99].

Job market challenges: The rapidly changing engineering and IT job market presents obstacles for graduates who may fail to match changing skill requirements [100].

Soft skills studies have a limited scope: Researchers emphasized the limitations of a Hong Kong study that examined only five soft skills for graduate students in the technology field [92].

Need for co-curricular courses: According to researchers, co-curricular courses focus on communication, which can improve teamwork, problem-solving, and leadership skills [101].

Integration of cognitive and emotional skills: To satisfy industry requirements, integrating social, spiritual, and emotional skills into the education curriculum is essential [102].

In summary, the difficulties in soft skills studies for IT students are caused by some variables, including communication difficulties, opposition to curriculum revision, the gaps between industry expectations and the education system, and unique challenges to the online teaching and learning process. Addressing these vital issues would require coordination among educational institutions, industry players, and policymakers.

Technical challenges: Distance learning involves technical and interpersonal difficulties [43,45]. Students often struggle with developing emotional skills, particularly self-regulation and motivation, in this context. Educators must incorporate activities and practices that foster emotional intelligence into their distance education programs.

### 4.3. The Suggestions Made for the IT Students’ Soft Skills Development

Several kinds of research on soft skills development for students in IT education provide useful recommendations. Caggiano et al. state that soft skills and cooperation training are necessary for collaboration tasks, especially in the engineering and science departments [82]. According to Lousa and Lousa, the impact of COVID-19 on students’ opinions of the Portuguese education system reveals lessons for policymakers [45]. Also, Siddo states that soft skills requirements differ across IT and non-IT businesses, with the former emphasizing technical skills and the latter stressing teamwork and personal attitudes [51]. Furthermore, Dowdall argues for implementing the MIMI methodology to improve computer science students’ collaboration and communication skills [62].

Project-based learning (PBL) is an important technique for developing soft skills. Dogara et al. discovered that PBL improves technical students’ teamwork, problem-solving, communication, and collaboration skills [59]. Also, Idris describes the e-colloquium as a platform for enhancing presentation, mental preparedness, and communication abilities [60]. Adamuthe and Patil suggest a roadmap for sophisticated problem-solving in CS/IT students, focusing on industry expertise and problem-solving methodologies [71].

Various learning methodologies are proposed to improve soft skills. Mitchell and Vaughan recommend team-based learning (TBL) for database classes to improve teamwork and performance [47]. On the other hand, Mgaiwa emphasizes the necessity of collaboration between universities and the IT industry in aligning education with industry demands [50]. To handle coursework loads and improve efficiency, mixing physical and mental activities into the curriculum is recommended [34].

Several technological platforms and paradigms assist the development of soft skills. The Passport online platform model allows for a self-reporting evaluation, improving IT industry-related soft skills and lifetime learning [77]. In the internship programs, the Rogaine Model minimizes gender gaps and promotes motivation. Gamification approaches and flipped classrooms are advocated for in terms of fostering engagement, motivation, and self-management abilities [44,88].

Several studies underline the importance of collaboration between academia and industry. Succi and Wieandt emphasize industry involvement in soft skills development [99]; however, Fitriani and Ajayi advocate for practical training programs to bridge the education–employment gap [69]. Internships and enterprise collaboration are important in improving graduate employability [94,103].

Said-Hung et al. highlights soft skills’ relevance in reconstructing social interactions and boosting university students’ performance in the post-pandemic educational landscape [43]. The Authentic Learning Scenario (ALS) approach is proposed to improve employability and develop a competitive academic profile [104].

Ultimately, these studies urge for incorporating long-term soft skills into IT education using a variety of methodologies such as (PBL), (TBL), industry engagement, technological platforms, and post-pandemic adaptability. They underline the importance of encouraging teamwork, communication, problem-solving, and adaptability, highlighting that these long-term skills are critical for students to thrive in the changing IT industry [77]. This viewpoint fits with the rising realization that a sustainable attitude and important soft skills are critical for handling modern difficulties and contributing to the IT sector’s long-term viability.

### 4.4. Emerging Soft Skills Trends in IT Education

This study reveals notable workforce development trends, such as a global emphasis on soft skills among IT students and engineers. A significant knowledge gap exists between graduate expertise and employer expectations. Therefore, the roles of industry participation and project-based learning in shaping HE programs to meet changing employment are critical.



Global Emphasis on Soft Skills: Engineers have excellent technical talents but struggle with social and communication skills. Fresh graduates struggle with cooperation, relationships, and expectations [82]. A clear knowledge gap occurs between graduate students’ expertise and employer expectations. The relevance of soft skills in addressing employment difficulties is emphasized [105]. Results are tailored to job requirements, highlighting personal knowledge, critical thinking, and innovation [67]. Soft skills development is influenced by remote learning, which includes technical challenges, learning circumstances, and interpersonal skills [45]. The HE institutions attempt to prepare Z-generation graduates equipped with hard skills. However, there is less attention to soft skills in technical and science courses such as engineering and IT [75].

Industry participation is critical in creating higher education programs to suit the changing demands of the employment market. A study by Siddo in Thailand noted that collaboration between IT experts and academics can considerably influence curriculum design, ultimately improving job opportunities for graduates [51]. In addition, this survey highlighted an awareness gap among educational institutions about industry requirements, underlining the necessity for stronger relations between academics and the industry. Similarly, Noah and Aziz’s study in Malaysia focused on enhancing the Teaching of English as a Second Language (TESL) program, which presented the importance of increasing graduates’ employability and soft skills [61]. The outcomes of this study provide significant insights for the reform of curricula in higher education institutions. This extreme shift implies that incorporating soft skills is important; nonetheless, it requires a gradual and well-planned transformation in legal education procedures.

Project-based learning is considered a driving force in strengthening students’ soft skills in the education system. A study by Dogara et al. in Nigeria highlights the reasonable influence of such learning on technical students’ soft skills, stressing communication, initiative, and problem-solving [106]. While emphasizing these advantages, the study discloses a surprising finding: Information and communication technology (ICT) disadvantages soft skill development. This viewpoint stresses the need to thoroughly recognize the factors influencing soft skills acquisition in project-based learning. These findings are useful for educators and policymakers seeking to improve learning methodologies for holistic skills development.

Several studies focused on the critical need for curriculum changes in higher education. Brennan conducted a study in Ireland highlighting the concern of employment shortage, underlining the importance of early incorporation of soft skills into curricula [46]. A Teaching of English as a Second Language (TESL) program in Malaysia is critical for better employability and soft skills [61]. Their observations can help with curriculum redesign in higher education institutions (HEIs). Concurrently, ref. [107], an Australian study, reveals redesign complexity difficulties, pointing to a worsening unemployment gap due to insufficient student and graduate industry engagement. The study advocates for a triangle collaboration among educators, industry, and students to successfully embed important soft skills into curricula, meeting the changing job market requirements. Moreover, in line with sustainability goals, including soft skills in curricula prepares students for urgent employment needs and adds to graduates’ long-term resilience and adaptability in a rapidly changing and sustainable job market. This comprehensive strategy assures that educational systems nurture technical proficiency and a long-term perspective for dealing with global concerns.

## 5. Discussion and Conclusions

This study demonstrated the literature on the soft skills approach in HEI in the IT field. The courses taught at universities require thorough investigation and new quality teaching techniques must overcome and replace traditional teaching. There is a high demand for students in the CS/IT field in the employment industry, especially after the industrial revolution 4.0. Unfortunately, graduates are well equipped in technical skills, not social aspects. The impact of hard skills has made it impossible to focus on the communication and management characteristics of the working environment. The fast development changes in the working industry and the industry revolution 4.0 have transformed the working environment rapidly and unexpectedly. This swift change in the working environment created a gap between fresh graduate students’ working industry environment. The literature has revealed that hard skills knowledge is inadequate to prepare students to enter the modern labor market. The current online teaching scenario has weakened students’ ability to connect and transform themselves into industry professionals. The issue of soft skills is widening the gap following the COVID-19 pandemic. New graduates with low soft skills are failing to make an impact in the new job market. The strategic design of pedagogy is essential in supporting students, despite the complexity of the model.

Resources beyond the capacity of a sole institute are required to solve this issue. In this case, all three stakeholders, namely, the universities, new graduates, and employers, must cooperate to bridge the gap already widened by the pandemic, as the modern workplace environment requires non-academic courses. However, additional practice during undergraduate study enhances the capability of students to be more competitive for employment. Several methods and models are mentioned in the literature to explain the issue of soft skills elements for HEIs and the working industry. The literature has found three vital aspects of soft skills: communication, teamwork, and problem-solving. These significantly empower IT graduates to access the new digital working environment in this IR 4.0 era. The HEIs must collaborate with the industry’s job requirements, because there are various forms of IT industry and non-IT industries concerning the soft skills requirements of employment. The MIMI method is beneficial for CS/IT students in developing teamwork and communication skills. In addition, emotional intelligence and soft skills must be developed when training future managers. In management positions in the IT field, the impact of emotional intelligence on leadership and decision-making skills is of great importance. In such courses, managers should be taught the skills to motivate their teams, communicate effectively in times of crisis, and develop empathy.

Also, running additional educational activities that support students in problem-solving issues is essential, along with other approaches mentioned in the literature. In addition, when training future managers, it is necessary to develop emotional intelligence and soft skills. In management positions in the IT field, the impact of emotional intelligence on leadership and decision-making skills is of great importance. In such courses, managers should be taught the skills to motivate their teams, communicate effectively in times of crisis, and develop empathy, including flipped-classroom combinations, game-based learning, problem-based learning, SSTP, and real problem-solving scenarios, adopting cognitive, social intelligence, and work-placement programs. Also, making arrangements in IT department curricula to develop emotional skills will enable learners to be more successful in their professional lives.

Two important factors raised in the literature are that students must be flexible and adaptable. Interestingly, the entrepreneurship scheme for university students before graduation by organizations is valuable for the smooth transition of graduates to employment. In return, the government is reducing the tax on selected companies offering students entrepreneurships.

Other notable points found in this systematic literature review are the lack of social justice and the digital divide in developing countries, which increase the unemployment rate of IT graduates.

Summarizing the systematic literature review results, soft skills are critical in information technology education. Soft skills shape student competencies and are one of the indications that they are not prepared for the labor market. According to the results obtained, leadership, problem-solving, socio-emotional skills, motivation, interpersonal communication, self-management and motivation soft skills are necessary for information technology students in many studies. Learners’ success in their future professional lives can be ensured by integrating the essential soft skills into the curriculum of educational institutions for employees with the qualifications that employers seek in the workforce. Another indication is difficulties such as the lack of interpersonal communication skills of learners in IT education, differences between industry expectations and the education provided in universities, difficulties in teamwork, difficulties caused by the gender variable, and the failure to integrate entrepreneurial skills into the curriculum. In addition, it has been revealed that education policymakers resist updating the curriculum, the industry blames universities for this issue, and the inadequacies in developing countries are prioritized over other skills instead of soft skills. These problems may cause learners to have difficulty adapting to their work lives and increase unemployment rates in the long term. Researchers may recommend that learners develop new learning–teaching models by addressing the challenges in developing soft skills.

Another research aim is to determine the suggestions in the literature for developing soft skills. The results underscore the importance of learning approaches such as project-based learning and collaborative learning. These approaches can create awareness for all stakeholders and foster constructive cooperation between universities and industry. The research also reveals that practical education applications can close the education–employment gap. Initiatives by policymakers to provide internship opportunities in the sector are also seen as a significant contributor. By systematically updating education policies and promoting the integration of soft skills into the education system, policymakers and practitioners can play a crucial role in providing a competitive advantage in the labor market and enhancing the flexibility capacity of the national workforce.

Another important result obtained is that Project-Based Learning is a learning approach that can significantly impact the development of many soft skills of learners. This finding should instill hope in educators and policymakers regarding this approach’s potential to shape the future of IT education. However, the factors hindering soft skill development must be analyzed well to implement this approach effectively.

Integrating sustainability skills into IT education programs, along with soft skills, can help learners understand environmental and social responsibility behaviors and develop individuals who promote sustainable practices in the business world.

It is worth mentioning that this review’s limitations have to do with the literature’s sample size. The articles were collected on two well-known database platforms, Web of Science and SCOPUS. However, some of the papers were not accessible due to fee charges. Also, the collected articles were limited because only two database platforms were used in this study. Additional sources such as IEEE Xplore, EBSCO, and other field databases can be included in future studies to provide more diversity in the systematic literature review.

Finally, this study highlights the importance of soft skills in IT education settings, illustrating diversity and interconnectedness. Despite the hurdles, the recommendations presented provide direction and guidance for educators, industry partners, and policymakers to work together to bridge the soft skills gap. As we mark the first anniversary of this inquiry, it serves as a reminder of the continuing relevance of soft skills in defining the future of IT professionals. The IT industry’s dynamic nature needs a continuing commitment to developing well-rounded, adaptive, and socially conscious professionals. This can be achieved through new educational methodology settings and collaborative initiatives.

This systematic literature review is a valid foundation for future studies. Moreover, the results of this study benefit both parties: the decision-makers at the university and the industry of the job market. The notes and the outcome of this study are vitally important in decreasing the unemployment rate among new graduates of IT and other scientific fields. In addition to that, according to the results, one of the important points raised in this study is cooperation, bridging the gap, and creating a swift transition of final-year students and graduates to the employment environment. For information technology industries to remain competitive in global markets, the soft skills deficiencies revealed in the studies must be eliminated. For this reason, higher education institutions and industries should cooperate to invest in the education system. Students studying information technology should be provided with the necessary professional and soft skills. Also, researchers should develop instructional designs that align with the results identified for sustainable education. Finally, this systematic literature review adds knowledge for further investigations on dynamic soft skills elements in the IT field.

The results of this study are expected to contribute to understanding the functionality and necessity of soft skills in the behavioral sciences literature. They are also expected to contribute to methodological innovations by illustrating how systematic literature review methods such as the PRISMA technique can be used effectively in behavioral science research.

## Figures and Tables

**Figure 1 behavsci-14-00894-f001:**
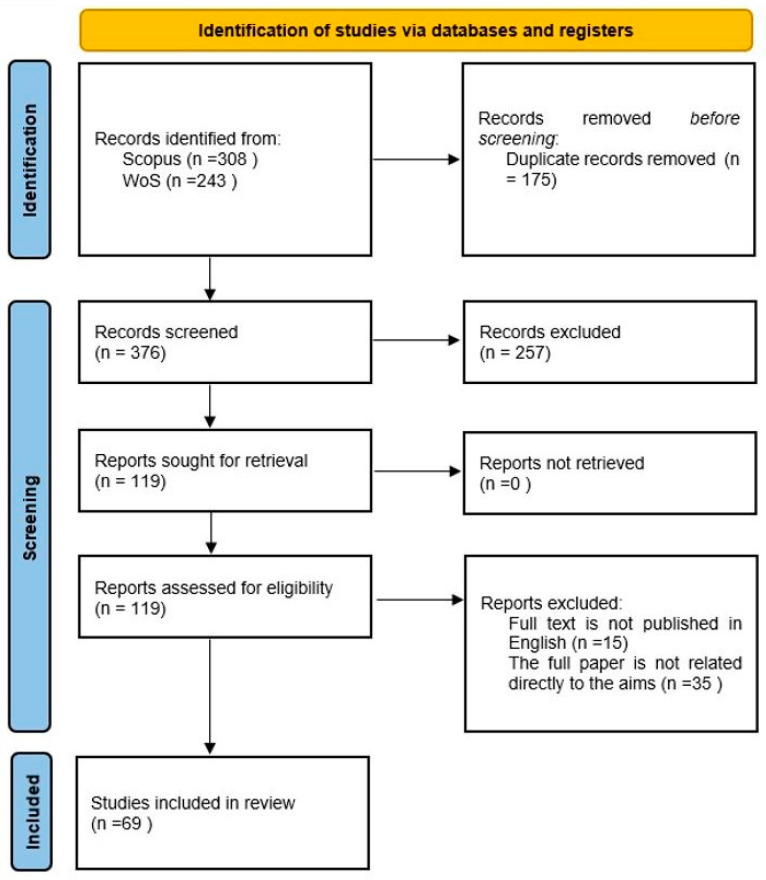
An illustration of the phases of the PRISMA diagram of a systematic literature review. Source: Page et al. [41].

**Table 1 behavsci-14-00894-t001:** Inclusion and exclusion criteria.

Characteristic	Inclusive	Exclusive
Context	Soft Skills in HEI	Documents which do not include soft skills in HEI
Type	Journals	Magazine
		Essays
		Reports
		Books
		Proceeding papers
Time	2018–2024	Before 2018
Language	English	Non-English
Availability	Open access	Access denied or chargeable

**Table 2 behavsci-14-00894-t002:** Themes related to soft skills and main results.

**Soft Skills Challenges and Gaps**	
[42]—Finland and Italy	IT engineering graduates have technical talents but struggle with social and communication skills. Cooperation, relationships, and expectations all present difficulties for recent graduates.
[43]—Spain	Remote learning affects soft skill development, including technological challenges, learning environments, and interpersonal skills.
[44]—Morocco	The relevance of soft skills is emphasized in the correlation between self-learning flipped classrooms and educational performance.
[45]—Portugal	As a result of COVID-19, remote learning has become a phenomenon, including technical difficulties, learning circumstances, and interpersonal skills.
[46]—Ireland	Concerns about job opportunities and the need for the early incorporation of soft skills into higher education curricula because of the recent pandemic.
[47]—Malaysia	Dominant elements of soft communication and teamwork skills are high and are a bonus for employment. However, there are weaknesses in other aspects of soft skills.
[48]—Scotland and USA	Soft skills elements are recommended for real-world professionalism, collaboration, and problem-solving. The teamwork produced interpersonal communication, dealing with frustration, and finding solutions.
[49]—24 countries	A curriculum design for IT courses must include teamwork and software project management. IT companies are unwilling to retrain new graduates.
[50]—Tanzania	Lack of cooperation between universities and industries. Regular educational curriculum reviews are a necessity for improving quality assurance.
**Industry-Specific Soft Skills**	
[51]—Thailand	IT expertise and academics can help to develop curricula to improve career chances. Educational institutions show a lack of awareness of industry requirements.
[52]—Portugal	Connect the development of hard and soft skills for success in informatics engineering. The emphasis is on creativity, self-evaluation, autonomy, and critical thinking.
[53]—Turkey	Top-tier cybersecurity courses focusing on soft skills for managerial roles are required. Tailoring education is important for industry demands.
[54]—Indonesia	Hard and soft skills must align with business skills. Lack of competence by new graduates in the IT field.
[55]—India	The new technology developments of AI, cloud computing, robotics, IoT, Big Data, and 3D printing are fewer skills without soft skills elements.
[56]—USA	The hard skills of technical experts are paradoxical to the interpersonal soft skills, habits, and personal quality. Ethics and communication skills are considered of high value in system design in an industry.
[57]—China	The company’s production in the working environment correlated with the performance of employees and employee satisfaction.
[58]—Spain	The value of soft skills in university education influences students’ success in entrepreneurship.
**Educational Approaches and Strategies**	
[59]—Nigeria	The soft skills of technical students are improved by project-based learning (PBL). Soft skills elements suffer as a result of ICT.
[60]—Indonesia	Online platforms and seminars support students in improving their psychological readiness and communication skills.
[61]—Malaysia	TESL program is perceived and developed for higher employability and soft skills. Altering the educational curriculum for (HEIs) is a necessity.
[62]—Poland and Ireland	Multinational, Intercultural, Multidisciplinary and Intensive (MIMI) technique is an attempt to improve teamwork in computer science courses.
[63]—Indonesia	Despite the strong correlation between these two elements, motivation is more effective than leadership for IT students in entrepreneurial aspects.
[47]—USA	Applying team-based learning (TBL) to leverage and increase soft skills. TBL improves conceptual and collaborative learning as teamwork scores are higher than individual work.
[34]—Ukraine	Physical education (PE) assists IT students in developing social quality skills, decision making, collaboration and teamwork. Physical activity classes support mental status and cognitive skills.
[64]—Slovakia	Soft skills have a significant effect on staff at academic centers of education. Self-determination by individual students that show the ability to communicate, interact, and solve issues. Academics and curriculum designers must know emotional self-regulation to cooperate and resolve conflicts.
[65]—Italy	The more involved students are with the university’s courses, the greater their desire to gain better academic careers.
[66]—Croatia	The neuromarketing approach using eye-tracking elements in online learning can predict students’ behavior and engagement process. It can improve certain skills of the behavior of students in HE institutions.
**Global Trends and General Soft Skills**	
[67]—Australia and UK	Results are tailored to job requirements, emphasizing personal knowledge, critical thinking, and innovation.
[68]—USA	It is a comparison of face-to-face versus synchronous approaches for teaching operating systems.
[69]—Malaysia	Interpersonal, intrapersonal, and business abilities are all required for work in the engineering field.
[70]—Israel	Passive methods have more of an effect on active methods. Matching different methods indicates different outcomes.
[71]—not specified	The roadmap model provides knowledge on designing curriculum-based complex problem-solving for CS/IT students.
[72]—Caribbean (Jamaica)	eMeetings for students to improve collaborative group-work assignments. Experience gained during student engagements in teamwork, communication, and problem-solving ultimately assists students in work placements.
[73]—Malaysia	The soft skills gap has raised alarms among industries of universities providing inexperienced workforce. Lack of communication, problem-solving, and teamwork creates a barrier for new graduates entering a job environment.
[74]—Malaysia	Communication, problem-solving, and teamwork have significant effects on soft skills integration. These elements improve employment chances.
[75]—USA, Germany and China	The demand for human skills by the industry is increasing because of the undersuppling of skilful science graduates for the job market.
**Technological Integration and Soft Skills**	
[76]—Italy	Soft skills integration into hardware and software solutions increases student capability.
[77]—Italy	Universities must engage in monitoring practices and develop learning to improve soft skills. Soft skills can help university transformation.
[78]—Malaysia	Educational institutions must provide further training on soft skills during uncertainty and complex situations such as pandemics. Training students on self-management skills.
[79]—Spain	The authors developed a practical framework in specific knowledge to motivate students in decision making using gamification principles. Motivational factors are the main key. Activity simulation in education adds a motivational value to knowledge and strengthens soft skills.
[80]—Italy	The concept of pedagogy transforms learners into leaders. Empowers student’s achievements in soft skills factors of creativity, cooperation, and redesigning projects.
[81]—Romania	SKILLS+ projects assist the eLearning process. Various soft skills techniques are useful in implementing ICT in the education sector and micro-firms to succeed.

## Data Availability

Data are contained within the article.

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
