# Peer review of "A Systematic Literature Review of Soft Skills in Information Technology Education"

_behavsci, 2024, doi:10.3390/bs14100894_

Round 1

Reviewer 1 Report

Comments and Suggestions for Authors

Suitable work for the journal, it is suggested to improve:

1. Define the research problem more precisely, updating the references.

2. Justify the research method

Comments on the Quality of English Language

Suitable work for the journal, it is suggested to improve:

1. Define the research problem more precisely, updating the references.

2. Justify the research method

Author Response

Reviewer 1

Our study was submitted for review under "A Systematic Literature Review Towards Soft Skills in Information Technology Education".

We thank you for your valuable feedback and suggestions. The updates we made regarding the issues you mentioned are stated below:

  1. Defining the Research Problem More Precisely: The introduction details the problem statement. This better reflects the aims and scope of our study and is also aligned with the current developments in the literature on the subject. We added new references in the relevant section that better explain the research problem and refer to the latest developments in the literature.

  1. Justifying the Research Method: We have detailed our research method. This explanation provides a comprehensive reason why our chosen methodology suits this study. In addition, the errors in the PRISMA flow diagram have been identified and updated.

Along with the above edits, we have made the necessary changes to the study's text and updated the relevant references. Thank you for contributing to the review process.

Important notice:

Reviewer 2 Report

Comments and Suggestions for Authors

Dear Authors,

Thank you for submitting your manuscript to the journal. I found your work engaging and insightful. However, there are some gaps that need to be addressed before a final decision can be made regarding its acceptance or rejection.

General Feedback:

  1. Introduction to Soft Skills:
    It is crucial to clearly define what soft skills are and explain their relevance to future employment or higher education students at the beginning of your manuscript. Without this foundational understanding, readers may struggle to fully grasp and appreciate the significance of your research.

  2. Motivation for Addressing Soft Skills:
    While you aim to explore soft skills, the rationale behind your focus on this topic needs to be clarified. Numerous studies have already examined soft skills in various contexts, so it’s important to articulate what motivated your research and how it adds to the existing body of knowledge.

  3. Literature Review and Practical Implications:
    I recommend including a dedicated section that reviews the current knowledge on soft skills, outlines your findings, and discusses the practical and policy implications of your research. This will provide a comprehensive context and enhance the applicability of your work.

  4. Methodology Clarity:
    The methodology, particularly the data analysis procedures, lacks clarity. It is difficult to assess the validity of your findings without a clear understanding of how the data was analyzed. Please provide a more detailed explanation of your methodology to ensure the credibility of your research.

  5. Justification for Systematic Review:
    The topic you are addressing is indeed important. However, you need to justify why you chose to conduct a systematic review rather than other approaches, such as a meta-analysis. This will help readers understand the methodological decisions behind your study.

  6. Discussion and Additional Sections:
    The discussion section is somewhat superficial and needs further development. Additionally, your manuscript is missing sections on limitations and practical implications. Please include these to provide a more rounded and impactful analysis of your findings.

By addressing these points, your manuscript will be significantly strengthened and better positioned for consideration in the journal.

Comments on the Quality of English Language

It is fine but can improve it for some issues such as long sentences, comma,...

Author Response

Reviewer 2

Our study was submitted for review under "A Systematic Literature Review Towards Soft Skills in Information Technology Education".

We thank you for your valuable feedback and suggestions. The updates we made regarding the issues you mentioned are stated below:

Introduction to Soft Skills:
It is crucial to clearly define what soft skills are and explain their relevance to future employment or higher education students at the beginning of your manuscript. Without this foundational understanding, readers may struggle to fully grasp and appreciate the significance of your research.

In the Introduction section, soft skills are explained, and their relation to the sector is explained with examples.

Motivation for Addressing Soft Skills:
While you aim to explore soft skills, the rationale behind your focus on this topic needs to be clarified. Numerous studies have already examined soft skills in various contexts, so it’s important to articulate what motivated your research and how it adds to the existing body of knowledge.

We expanded the introduction to more clearly articulate the motivation for our research. In particular, we detailed the reasons for our focus on soft skills and how we contribute to the existing literature on this topic.

Literature Review and Practical Implications:
I recommend including a dedicated section that reviews the current knowledge on soft skills, outlines your findings, and discusses the practical and policy implications of your research. This will provide a comprehensive context and enhance the applicability of your work.

In the "Discussion and Conclusion" section, we elaborate on the current state of the literature and our findings. We present the implications of the results for practice and policy in more detail.

Methodology Clarity:
The methodology, particularly the data analysis procedures, lacks clarity. It is difficult to assess the validity of your findings without a clear understanding of how the data was analyzed. Please provide a more detailed explanation of your methodology to ensure the credibility of your research.

We have detailed our research method, which provides a comprehensive explanation of why our chosen methodology suits this study. In addition, the errors in the PRISMA flow diagram have been identified and updated.

Justification for Systematic Review:
The topic you are addressing is indeed important. However, you need to justify why you chose to conduct a systematic review rather than other approaches, such as a meta-analysis. This will help readers understand the methodological decisions behind your study.

In the method section of the study, it is explained why systematic literature review was chosen and other methods were not chosen.

Discussion and Additional Sections:
The discussion section is somewhat superficial and needs further development. Additionally, your manuscript is missing sections on limitations and practical implications. Please include these to provide a more rounded and impactful analysis of your findings.

In the discussion and conclusion section, the practical implications and limitations of the study are discussed.

Along with the above edits, we have made the necessary changes to the study's text and updated the relevant references. Thank you for contributing to the review process.

Important notice:

Additions made based on your Green highlights suggestions.

Updates made based on your Yellow highlights suggestions

Best regards,

Reviewer 3 Report

Comments and Suggestions for Authors

**This study aims to investigate the role of soft skills in higher education in information technology (IT) and assess their impact on the employability of graduates in the IT industry. The study seeks to identify trends in soft skills elements within studies conducted for IT students and how these skills can bridge the gap between industry expectations and the technical capabilities of graduates. Ultimately, the study aims to contribute to the understanding of the functionality and necessity of soft skills in the behavioural sciences literature, particularly in the context of IT education.**

**However, there are certain reasons that lead me to reject the manuscript in its original version:**

1. **Lack of originality:** The article does not present a novel contribution to the field of behavioural sciences applied to education in information technology. The systematic review it conducts is based on well-known studies and does not offer new perspectives or conclusions that add value to the existing knowledge. New perspectives should be addressed.

2. **Methodological inconsistency:** Although the article mentions the use of the PRISMA methodology for the systematic review, the description of the process is ambiguous and lacks details on how the inclusion and exclusion criteria were applied, which questions the validity and reliability of the results obtained.

3. **Limited source selection:** The study is based solely on two databases (SCOPUS and Web of Science), which could have limited the diversity and breadth of the studies considered. This may lead to bias in the review, as other relevant studies may have been excluded. It is necessary to explain why other databases were not selected.

4. **Structural and writing issues:** The article presents problems in the organisation of ideas and clarity in the exposition. There are sections where the writing is confusing, and some arguments are not well-founded, which hinders the understanding of the key points intended to be communicated. There are also translation issues, as well as words in Spanish.

5. **Lack of critical review:** The article does not provide a critical discussion of the limitations of the reviewed studies or the implications of the findings in educational practice. This limits the article's usefulness for researchers and professionals looking to apply this knowledge in real-world contexts.

6. **Citation and reference problems:** Some studies mentioned in the article are not adequately cited, or the references are incomplete, which affects the academic credibility of the work.

7. **Weak conclusions:** The article's conclusions are generalised and not well supported by the data presented. The connection between the findings and practical recommendations is vague, diminishing the potential impact of the article on the academic community.

**For these reasons, I recommend the rejection of the article in its current form, suggesting that the authors address these issues before resubmitting it.**

Author Response

 Reviewer 3

Our study was submitted for review under "A Systematic Literature Review Towards Soft Skills in Information Technology Education".

We deeply appreciate your insightful feedback and suggestions. Your input has been instrumental in shaping the updates we made regarding the issues you mentioned, which are stated below:

  1. **Lack of originality:** The article does not present a novel contribution to the field of behavioural sciences applied to education in information technology. The systematic review it conducts is based on well-known studies and does not offer new perspectives or conclusions that add value to the existing knowledge. New perspectives should be addressed.

 We have extensively revised the introduction, method, findings, discussion and conclusion sections. In line with these revisions, we have updated our study to reflect new perspectives and current developments, along with existing information in the literature. In particular, we have more clearly stated how this study can contribute to the field. With these revisions, we hope that our study will make a more original and valuable contribution to the field of behavioral sciences and information technology education.

  1. **Methodological inconsistency:** Although the article mentions the use of the PRISMA methodology for the systematic review, the description of the process is ambiguous and lacks details on how the inclusion and exclusion criteria were applied, which questions the validity and reliability of the results obtained.

We have detailed our research method, which comprehensively explains why our chosen methodology suits this study. In addition, the errors in the PRISMA flow diagram have been identified and updated.

  1. **Limited source selection:** The study is based solely on two databases (SCOPUS and Web of Science), which could have limited the diversity and breadth of the studies considered. This may lead to bias in the review, as other relevant studies may have been excluded. It is necessary to explain why other databases were not selected.

We explained why our study used only SCOPUS and Web of Science databases. This information explains why other databases were not selected and is included in the method section. In the conclusion section, we stated that the databases used were among the study's limitations. We also suggested the advantages of using a wider database and possible improvements. With these adjustments, we expressed the methodological limitations of our study more clearly and made suggestions for future research.

  1. **Structural and writing issues:** The article presents problems in the organisation of ideas and clarity in the exposition. There are sections where the writing is confusing, and some arguments are not well-founded, which hinders the understanding of the key points intended to be communicated. There are also translation issues, as well as words in Spanish.

Thank you for your suggestions. The entire study was reviewed.

  1. **Citation and reference problems:** Some studies mentioned in the article are not adequately cited, or the references are incomplete, which affects the academic credibility of the work.

All references were edited with Mendeley software to ensure that there were no omissions or errors. 

  1. **Weak conclusions:** The article's conclusions are generalised and not well supported by the data presented. The connection between the findings and practical recommendations is vague, diminishing the potential impact of the article on the academic community.

In the discussion and conclusion section, the practical implications and limitations of the study are discussed. Also, discussions based on the results were made.

Along with the above edits, we have made the necessary changes to the study's text and updated the relevant references. Thank you for contributing to the review process.

Important notice:

Additions made based on your Green highlights suggestions.

Updates made based on your Yellow highlights suggestions

Best regards,

Round 2

Reviewer 2 Report

Comments and Suggestions for Authors

Dear authors

thank you for resubmitting your manuscript after revision. Now it is more clear and acceptable to be consider for publication in the MDPI journal. You did great job in the revision version. Again thank you

Author Response

Thank you very much for your feedback and valuable contributions. We are very happy that you liked the revised version of our work. Your feedback has greatly contributed to making our article better. Thank you again for your interest and support.